# Quality of Life Evaluation Using SRS-30 Score for Operated Children and Adolescent Idiopathic Scoliosis

**DOI:** 10.3390/medicina58050674

**Published:** 2022-05-18

**Authors:** Alexandru Herdea, Teodor Alexandru Stancu, Alexandru Ulici, Claudiu N. Lungu, Mihai-Codrut Dragomirescu, Adham Charkaoui

**Affiliations:** 111th Department of Pediatric Orthopedics, “Carol Davila” University of Medicine and Pharmacy, 050474 Bucharest, Romania; alexherdea@yahoo.com; 2Pediatric Orthopedics Department, “Grigore Alexandrescu” Children’s Emergency Hospital, 011743 Bucharest, Romania; mcodrutd@gmail.com; 3General Medicine Department, “Carol Davila” University of Medicine and Pharmacy, 050474 Bucharest, Romania; teodor.stancu95@gmail.com; 4Department of Surgery, Country Emergency Hospital Braila, 810249 Braila, Romania; lunguclaudiu5555@gmail.com; 5Department of Morphological and Functional Sciences, Faculty of Medicine and Pharmacy, “Dunărea de Jos” University of Galați, 800008 Galați, Romania; charkaoui.adham@gmail.com

**Keywords:** adolescent idiopathic scoliosis (A.I.S.), spinal deformity, adolescent, SRS-30 questionnaire, Cobb angle, quality of life, patient self-image

## Abstract

*Background and objective*: Adolescent idiopathic scoliosis (A.I.S.) is a disorder with a significant impact on health and self-image. This spinal deformity can affect between 2% and 4% of the adolescent population and may alter one’s quality of life. This study aims to assess the patient outcome, satisfaction, and quality of life following surgical treatment using the SRS-30 questionnaire. *Materials and Methods*: A number of 49 children and adolescent patients diagnosed with idiopathic scoliosis that had surgery were included in this study. They thoroughly completed the SRS-30 questionnaire before and after the surgery, based on which data analysis was carried on. Correlations between the test results and imagistic data (pre- and postoperative Cobb angle, correction rate of Cobb angle, number of instrumented spinal segments, and number of pedicle screws/laminar hooks used in the surgery) were performed. *Results:* Our results showed that 87.76% of the patients were girls, and the mean age at surgery was 14.83 years. Postoperatively, the Cobb angle improved significantly (*p* < 0.0001). The questionnaire domain “Satisfaction with management” improved dramatically after surgery, averaging 13.65 points (91% out of the maximum score). The average postoperative test score was 125.1 points. Statistically significant correlations were found between the correction rate and SRS-30 score improvement (*p* < 0.001), in total as well as per each domain of the survey, respectively. Comparing the questionnaire domains, “Self-image” was positively correlated with “Satisfaction with management” (*p* < 0.0001). *Conclusions*: Better correction rate led to higher values of SRS-30 score. Additionally, the younger the age at surgery is, the higher the score. The number of instrumented spinal segments does not alter the quality of life. Overall, the most crucial factor influencing patient satisfaction after surgical treatment is self-image.

## 1. Introduction

Adolescent idiopathic scoliosis (A.I.S.) is a three-dimensional (3D) deformity of the spine and trunk that mainly affects previously healthy children, concerning 2–4% of the adolescent population [1]. Scoliosis can also be diagnosed at very young ages, and the management can be complex.

Scoliosis is a disease with an insidious onset, in which the deformity of the spine and less or not at all back pain predominate. It is a condition that is found in 2–4% of children and adolescents, and the genetic factor (inheritance) is often incriminated. The first signs are given by the asymmetry of the shoulders and/or the asymmetry of the pelvis, and as the disease progresses, these asymmetries progress and there is evidence of a hump on a hemithorax due to vertebral rotation. In severe cases, respiratory obstruction, pain, and comorbidities can occur. Being a progressive disease, in the case of severe scoliosis, it will continue to deform the spine and show symptoms in adult life. The Cobb angle is a marker of staging and tracking evolution, calculated at the level of the curvature of the spine (or curvature if there is more than one). The treatment of idiopathic scoliosis is performed by physiotherapy for cases up to 25 degrees, physiotherapy and bracing for a Cobb angle between 25–45°, and surgery is recommended when the threshold of 45 degrees is exceeded.

Scoliosis management comprises physiotherapy for mild cases, bracing for moderate scoliosis, and surgery for Cobb angles of 45–55° or more, respectively [2]. In contrast to the impact on quality of life, A.I.S. also has an appreciable economic effect, as the treatment approaches are mainly costly [3,4].

This study aimed to assess patient outcome and Satisfaction following posterior vertebral instrumentation and fusion using the Scoliosis Research Society 30 (SRS-30) questionnaire. In Romania such studies were never conducted, even if spinal surgery was done, ever since 1980. One of the aims of this study was to compare our results with the literature.

Surgical treatment may be necessary to improve the aesthetic appearance, self-image, and lung function, and control the curve progression [5]. Additionally, it has been shown that a successful surgery is not always correlated with subjective patient satisfaction, as patients and doctors perceive the results differently [6,7].

Due to the severe impact of A.I.S. on quality of life, several authors tried to evaluate the success of the treatment, and, in achieving this, SRS-30 questionnaires have proven to be helpful [8,9,10,11].

We chose to use the SRS-30 questionnaire instead of its variants (SRS-22 or SRS-24) because this version includes 7 post-surgery questions. Other questionnaires have also been used to evaluate adolescent idiopathic scoliosis surgical treatment outcomes, such as the SRS-24, SRS-22, SF-12, or SF-36, but most studies that used these surveys were retrospective. Therefore, these questionnaires need to be administered prospectively, before and after treatment, to obtain accurate Health-related quality of life (HRQoL) measurements.

SRS-30 is a practical tool developed to measure the outcomes of surgical treatment in idiopathic scoliosis. It is a Patient Outcome Questionnaires released by Scoliosis Research Society. It was recently introduced as a combination of SRS-24 and SRS-22, being more complex and better formulated than the previous versions.

As other authors stated, the risk of complications for the spine surgery is varying but with a high rate, thus quality of life evaluation should be considered mandatory in order to improve outcomes for future patients [12].

To determine the impact of disease on patients, we evaluated “health-related quality of life” (HRQoL) [13,14], using the Scoliosis Research Society 30 questionnaire (SRS-30) together with preoperative and postoperative radiographs in operated children and adolescent idiopathic scoliosis.

## 2. Materials and Methods

We established a prospective cohort study performed in the Pediatric Orthopedics Section of a children’s hospital located in an urban area between the years 2011 and 2018. The approval of The Ethics Committee from “Grigore Alexandrescu” Children’s Emergency Hospital, Bucharest, Romania was obtained before the initiation of the presented study. The protocol code is 24 and date of approval is 11 March 2011.

Inclusion criteria were the following: patients diagnosed with children and adolescent idiopathic scoliosis proposed for surgery by a posterior approach using a hybrid technique of posterior vertebral fusion and segmental instrumentation with pedicle screws, laminar hooks, and 2 rods.

Exclusion criteria were as follows: non-idiopathic scoliosis, absence of preoperative and postoperative radiographs, different surgical approaches, associated comorbidities, a follow-up of less than 18 months, lack of consent to enroll in the study, incomplete answers to the questionnaire.

The interventions were performed by a specialized team with experience in the surgical treatment of scoliosis. General anesthesia was used to perform surgery, and the procedures had an average time of 6 h. Patients remained hospitalized on average for 7 days postoperatively until they could stand up and resume walking.

Patients’ quality of life was evaluated before and after scoliosis surgery. SRS-30 questionnaire and data recordings from each patient were stored in electronic format. Preoperative and postoperative radiographs were analyzed. After the patients were proposed for surgery, they were invited to participate in the survey.

The SRS-30 questionnaire consists of 30 questions, each with three to five answer choices. The questionnaire was translated and validated by the English Department from our Faculty of Medicine. The score sheet divides the survey into five domains: “Self-image/Appearance” (45 points), “Function/Activity” (35 points), “Pain” (30 points), “Mental health” (25 points), and “Satisfaction with management” (15 points). Higher scores show a better quality of life.

With the help of preoperative radiographs, patients were divided according to the Lenke classification. This triad system classification consists of curve type, lumbar spine modifier, and sagittal thoracic modifier. This classification is comprehensive; all types of curves can be described as two-dimensional, with an increased emphasis on the sagittal plane, and reliable, having excellent inter and inner observer reproducibility. Furthermore, it is reliable, objective, and practical.

In addition, preoperative and postoperative Cobb angles, correction rate of Cobb angle, number of instrumented spinal segments, and number of pedicle screws/laminar hooks were valuable data used in the study. Student T-test was used to analyze these data.

Pearson Correlation coefficient test was used to highlight correlations between age and correction rate, correction rate, and SRS-30 total score, as well as divided in each domain. The same test was used to assess correlations between independent domains of the questionnaire. Statistical analysis was performed using I.B.M. S.P.S.S. statistics V22.0 (IBM SPSS Inc, Chicago, IL, USA), and a *p*-value < 0.05 was considered statistically significant. 

The patients were divided into three class intervals as follows: 9–11 years, 12–14 years, and 15–17 years, respectively.

Finally, Lenke’s classification is discussed. The Lenke classification system is designed with respect to curve type (1–6) with a sagittal thoracic modifier (−, N, or +) and a lumbar spine modifier (A, B, C). The Scoliosis Research Society (S.R.S.) established all definitions for the curve types. Structural curves are defined as at least +25° on bending films or >+20° of kyphosis. When multiple curves are present, the significant curve is defined as the most considerable curve; however, mild curves are defined as structural. The basis of the system is to determine which vertebral levels were appropriate for arthrodesis. Curve types are based on location. Thoracic curves possess an apex from T2 to the T11/T12 disc. Thoracolumbar curves present an apex at T12 to L1, and lumbar curves have an apex from the L1/L2 disc to L4. Type 1 is a primary thoracic curve. There are also proximal thoracic and thoracolumbar/lumbar nonstructural mild curves. Type 2 is a double thoracic curve with a proximal minor structural curve and a primary thoracic significant curve, respectively. Type 3 is also a double significant curve, with the primary thoracic curve being the major curve. The second curve is in the lumbar region, and both curves are structural. Type 4 is a triple significant curve with structural curves in the proximal thoracic, main thoracic, and TL/L regions. Once again, the thoracic curve is the significant curve. Type 5 is a single structural curve in the thoracolumbar/lumbar region surrounded by two minor nonstructural curves. Type 6 is a thoracolumbar/lumbar-main thoracic double curve. The thoracolumbar/lumbar is the significant curve, but the thoracic curve is also structural.

## 3. Results

Forty-nine patients were included in the study—43 girls (87.76%) with a mean age of 14.74 ± 1.47 years, and 6 boys (12.24%) with a mean age of 15.5 ± 1.71 years.

The questionnaire was completed on average two days before the surgery and 4.32 years after the surgery. Time between surgery and postoperative survey completion varied between 1.5 years and 10 years. Patient follow-up was every 6 months for at least 2 years. Postoperative data evaluation was done in 2018 when the patients came for the regular 6 months follow-up, thus having the postoperative evaluation done at a mean of 4.32 years after surgery. The mean age of patients completing SRS-30 postoperatively was 19.63 years with a minimum of 14 years and a maximum of 25 years. Regardless of the time after surgery, each questionnaire domain was thoroughly explained to the patient, and there were no differences in patients’ perspectives on the survey, as SRS-30 questions are independent of age or socioeconomic status.

According to the Lenke classification, most children (25, 51.02%) were identified with a Lenke type 1 scoliosis. In addition, there were 8 (16.33%) patients with a type 2 curve, 3 (6.12%) with type 3, 3 (6.12%) with type 4 curve, 4 (8.16%) with type 5, and 6 (12.24%) patients with Lenke 6 Thoracolumbar/Lumbar Main Thoracic curvature.

Following the surgery, the Cobb angle improved significantly (*p* < 0.0001), with an absolute difference of 36.5° and an average correction rate of 60.11% (Figure 1).

Regarding the preoperative SRS-30 questionnaire, the lowest scores were in the field of “Self-image/Appearance”, with a mean of 15.66 points (out of 30 points). On the other hand, the mean total score was 74.4 points out of 115 (64.69% of the maximum score) (Table 1).

Regarding the SRS-30 questionnaire completed after the surgery, patients had the highest scores in the field evaluating “satisfaction with management” with an average of 13.65 points (out of 15 points), meaning 91% of the maximum score. The other domains (function/activity, pain, self-image/appearance, and mental health) also scored better, ranging from 80% to 85% of the maximum score. The mean total score was 125.1 points (83%) (Table 2).

A comparison between preoperative and postoperative scores revealed statistically significant improvement in all domains (*p* < 0.05) (Table 3). 

Patients identified with a Lenke type 6 curve had the highest correction rate (65.34%), while children with Lenke type 3 scoliosis had the lowest (48.66%). However, the differences were not statistically significant (*p* = 0.0569) (Figure 2).

No statistically significant gender-related differences in the SRS-30 scores were found. In male patients, the function/activity mean was 29, the pain was 24.67, self-image/appearance was 37.5, mental health was 10.17, satisfaction with management was 13.33, and the total mean was 123.67, respectively. In female patients, the function/activity mean was 28.3, the pain was 25.74, self-image/appearance was 37.46, mental health was 20.09, satisfaction with management was 13.69, and the total mean was 125.3, respectively. As stated before, correlation between those variables is statistically insignificant, with *p* = 0.4912.

The correction rate correlates with postoperative survey domains ”Self-image/Appearance” (R = 0.3235, *p* = 0.0233), ”Satisfaction with management” (R = 0.3751, *p* = 0.0079), and also with the total SRS-30 score (R = 0.3470, *p* = 0.0143) (Table 4).

Correlations between: ”Function/activity” and ”Pain” (R = 0.4641, *p* = 0.007), ”Function/activity” and ”Self-image/Appearance” (R = 0.6499, *p* < 0.0001), ”Function/activity” and ”Mental health” (R = 0.4494, *p* = 0.0011), ”Function/activity” and ”Satisfaction with management” (R = 0.6555, *p* < 0.0001), ”Pain” and ”Self-image/Appearance” (R = 0.4442, *p* = 0.0013), ”Pain” and ”Mental health” (R = 0.5625, *p* = 0.0002), ”Pain” and ”Satisfaction with management” (R = 0.5219, *p* = 0.00012), ”Self-image/Appearance” and ”Mental health” (R = 0.3527, *p* = 0.12933), ”Self-image/Appearance” and ”Satisfaction with management” (R = 0.8079, *p* < 0.0001) are also to be noted.

Preoperative and postoperative SRS-30 total scores grouped by age intervals and domains can be seen in Table 4.

A negative correlation between age and postoperative scores was observed. Age also negatively correlated with “Self-image/Appearance” (R = −0.30, *p* = 0.03) and “Mental health” (R = −0.32, *p* = 0.02) individually (Table 5).

However, no statistical significance was found when comparing the number of instrumented spinal segments and the SRS-30 domains that evaluate function/activity and pain. Regarding the function/activity domain for the number of instrumental spinal segments, the correlation was R = 0.0971 (*p* = 0.5068) and R = 0.1366 (*p* = 0.349) for the number of pedicle screws and laminar hooks, respectively. On the other hand, the pain domain showed an R = 0.0835 (*p* = 0.5684) for the number of instrumental spinal segments and R = 0.0165 (*p* = 0.910) for the number of pedicle screws laminar hooks, respectively.

## 4. Discussion

Children and adolescent idiopathic scoliosis is a disorder diagnosed based on clinical examination and imagistic study (X-rays). Girls are more likely to suffer from this deformity (80% of total cases) [15]. Adolescence is a period of personal development in which children become aware of their self-image and build confidence and self-respect [16]. These aspects can make life difficult for patients with A.I.S. and may lead to mental disorders even if it is not a life-threatening disease [17].

Pellegrino et al. concluded that surgical treatment of adolescent idiopathic scoliosis was associated with quality of life improvement, as measured with the SRS-30 questionnaire. However, in his study, postoperative quality of life was not correlated with curve magnitude (Cobb angle), correction rate, or type of instrumentation [18].

Ghandehari et al. also found that the correction rate can significantly alter the overall score of SRS-30 and that it is positively correlated with patients’ “Satisfaction with management”. Regarding individual domains of the questionnaire, a positive correlation was found between “Self-image/Appearance” and “Satisfaction with management” [11]. 

Ng et al. found gender-related differences in preoperative and postoperative scores within different questionnaire domains. Statistically significant correlations were highlighted in female patients for the domains assessing pain and satisfaction with management after the surgery and male patients for the function/activity domain. It was also noted that high preoperative values of Cobb angle and high correction rates were two significant factors that influenced “Self-image” and “Satisfaction with management”, especially in male patients [19].

In addition to the results of the studies mentioned above, we found a positive correlation between correction rate and “Self-image/Appearance” and a negative correlation between age and total score of the SRS-30 survey, so younger age at surgery is associated with quality of life improvement. Additionally, we found that individual domains of the questionnaire correlate with each other. It was noted that the correlation between “Self-image” and “satisfaction with management” had the most important statistical significance (R = 0.8079, *p* < 0.0001).

Furthermore, in a meta-analysis, Zeng et al. indicated that brace-treated A.I.S. patients had a higher QoL. Further analysis could not be performed due to insufficient data, and the authors were unable to make a proper analysis of QoL for different types of A.I.S. and the therapeutic methods chosen by brace-treated A.I.S. patients [20]. In an extensive study of 1.205 pupils with idiopathic scoliosis, Rainoldy et al. showed that all other scores were greater than 4 points with small standard deviations. Females showed differences among groups for all domains and the total score (*p* < 0.05). Function, pain, and mental health in males did not show statistically significant differences among groups (*p* > 0.1). All differences found were minimal (0.5 points). The correlations with the severity of deformity measures were low (rs < 0.289). Moore, according to the results, the deformity is not a real issue for A.I.S. before a diagnosis is made, treatment plan, and specialists interfere with their everyday life. Scoliosis Research Society 22 Questionnaire showed some discriminative validity between small and large curves, but the differences found were small [21]. Additionally, Ng et al., in a study from 2009–2013 regarding HRQoL using S.R.S. 30 score of patients with idiomatic scoliosis, found that the mean age was 16.28 at surgery, and 83.6% were female. Important correlations between pre-op scores and scores after surgery were observed. No gender difference in all 5 domain scores at the 3-time points was found. Pre-op maximum Cobb angle and corrected angle were demonstrated to be risk factors for self-image and satisfaction with management in male and female patients. The authors conclude that medical professionals should pay attention to the difference in personal perceptions of feelings between boys and girls. Special care should also be allocated to A.I.S. patients and arrange earlier surgical intervention [19].

Regarding the SRS-30 scores obtained in our study, there were no gender-related statistically significant differences.

There were no correlations between the number of instrumented spinal segments or the number of pedicle screws/laminar hooks and the domains that evaluate function/activity and pain.

Some study limitations are noted, such as a lack of statistical correlations between Lenke’s classification and SRS-30 score, lack of vertebral rotation measurements (Nash and Moe, for example), and an unbalanced male–female ratio. For a powerful impact study, further studies are to be done to assess a larger population size that may encompass more male patients, with correlations between Lenke’s classification and SRS-30 score and vertebral rotation measurements.

The answers to the postoperative questionnaire are perfectly valid both at 1.5 years and 10 years after the surgery because at that age (mean age of 14.74 years), the body is sufficiently mature, and the patient is aware enough to answer all of the survey questions that target day-to-day aspects of life.

The test’s statistical power is 0.9747213 computed for *p* = 0.5, α error probability = 0.05, total sample size = 48 non centrality parameter ϕ = 4.0, critical *t* = 2.0128 (Figure 3). The prediction of power (1-β error probability) plotted on Y, and the total sample size plotted on x, is represented in Figure 3.

## 5. Conclusions

A greater correction rate of the Cobb angle is associated with increased patient satisfaction and better quality of life improvement (HRQoL).

The increase in the total SRS-30 score is remarkable, as the surgery is performed at a younger age. This must be taken into account by doctors, patients, and their families as early as the diagnosis if surgery is required. 

Minimizing the number of pedicle screws or laminar hooks should not be a priority of preoperative planning because it does not alter the patient’s quality of life.

The most important factor influencing patient satisfaction after surgical treatment is self-image.

## Figures and Tables

**Figure 1 medicina-58-00674-f001:**
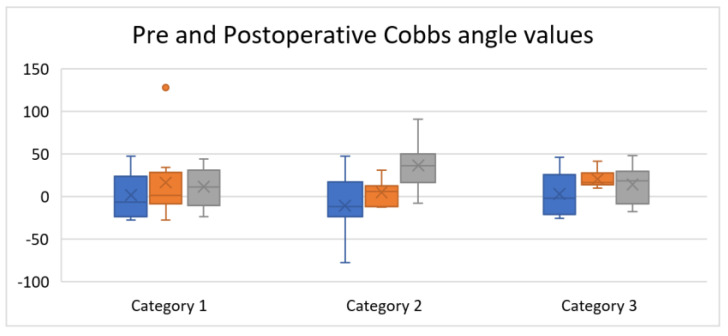
Comparison between the pre- and postoperative Cobb angle values. Confidence interval of 95% (CI 95%) for preoperative Cobb angle was between 57.8–65.9°, for postoperative Cobb angle, was between 21.8–28.8°, the absolute difference was between 33.9–39.3°, and the correction rate was between 56.1–64.1%, respectively. *p*-value was <0.0001. Category 1-Mean value, Category 2-Standard deviation, Category 3-CI (95%).

**Figure 2 medicina-58-00674-f002:**
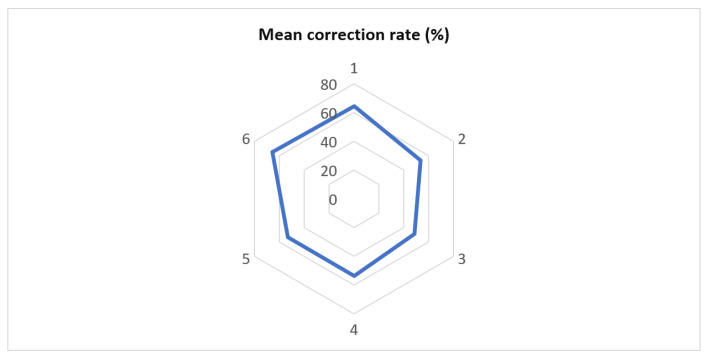
The average percentage of correction compared to different groups of Lenke’s classification.

**Figure 3 medicina-58-00674-f003:**
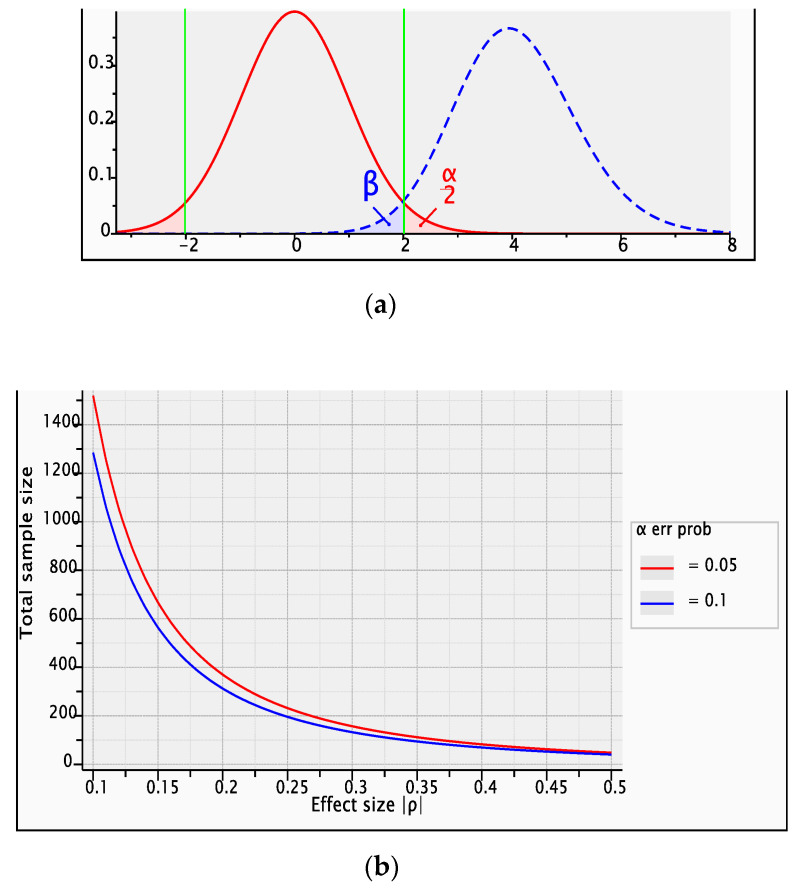
Computed statistical power (**a**) and validation by prediction of study power (**b**) by sample size (*n* = 48) with β probability of type II error (false negative) and α probability of type I error (false positive).

**Table 1 medicina-58-00674-t001:** The mean score, percentage of the maximum score, and confidence interval of Scoliosis Research Society 3 (SRS-30) and its domains before the surgery.

	Mean (Points)	Percentage Out of the Maximum Score	Confidence Interval (CI = 95%)
Function/Activity	18	60%	17.1–18.9
Pain	17.46	58.2%	16.8–18.1
Self-image/Appearance	15.66	44.74%	15.1–16.2
Mental health	14.2	47.33%	13.8–14.6
Satisfaction with management	8.5	85%	8.3–8.7
TOTAL	74.4	64.69%	70.2–78.6
Average points per question	3.23	64.69%	3.09–3.37

**Table 2 medicina-58-00674-t002:** The mean score, percentage of the maximum score, and confidence interval of SRS-30 and its domains after the surgery.

	Mean (Points)	Percentage Out of the Maximum Score	Confidence Interval (CI = 95%)
Function/Activity	28.38	81%	27.3–29.5
Pain	25.61	85%	24.6–26.6
Self-image/Appearance	37.46	83%	36.2–38.8
Mental health	19.97	80%	19–21
Satisfaction with management	13.65	91%	13.1–14.2
TOTAL	125.1	83%	121–129
Average points per question	4.17	83%	4.04–4.3

**Table 3 medicina-58-00674-t003:** Comparison between preoperative and postoperative SRS-30 scores, using the Pearson Correlation coefficient test with *p* value < 0.001.

SRS-30 Domains	Preoperative Score (Mean Points per Question)	Postoperative Score (Mean Points per Question)	*p*-Value
Pain	3.49	4.26	<0.001
Self-image	2.61	4.16	<0.001
Function/Activity	3.6	4.05	<0.001
Mental health	2.84	3.99	<0.001
Satisfaction with management	4.25	4.55	<0.001
Total	3.23	4.17	<0.001

**Table 4 medicina-58-00674-t004:** Mean scores of SRS-30 questionnaire by age groups before and after the surgery.

	Preoperative SRS-30 Scores			Postoperative SRS-30 Scores		
Age group (years)	9–11	12–14	15–17	9–11	12–14	15–17
Pain	19.11	17.21	15.89	28.86	25.81	23.89
Self-image	27.67	25.21	22.67	42.33	37.31	34.77
Function/Activity	20.89	18.93	17.76	30.86	28.93	26.11
Mental health	14.67	13.82	13.06	22.5	21.37	17.77
Satisfaction with management	10.11	9.33	8.67	14.87	13.66	12.67
Total	92.45	84.5	78.05	139.42	127.08	115.21

**Table 5 medicina-58-00674-t005:** Correlations between patient’s age, total SRS-30 score, and its domains.

	Correlation with Age	
Function/Activity	R = −0.1507	*p* = 0.3036
Pain	R = −0.1171	*p* = 0.4233
Self-image/Appearance	R = −0.3018	*p* = 0.0355
Mental health	R = −0.3213	*p* = 0.0245
Satisfaction with management	R = −0.2136	*p* = 0.1405
TOTAL	R = −0.2852	*p* = 0.0471

## Data Availability

All of the data are registered at “Grigore Alexandrescu” Children’s Emergency Hospital, Bucharest, Romania.

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
