# Peer review of "Quality of Life Evaluation Using SRS-30 Score for Operated Children and Adolescent Idiopathic Scoliosis"

_medicina, 2022, doi:10.3390/medicina58050674_

Round 1

Reviewer 1 Report

Respected Authors,

I am glad that you have done some corrections. However, the updated manuscript still lacks essential information. Need for the study, validation on questionnaire translation, details on follow-up data collection, and support for the conclusion are not provided or addressed.

I feel that the following points were not properly addressed or discussed in the manuscript.

  1. The need for conducting this study was not properly discussed and justified. There are several other papers which are already been published. How far this paper is different from others?
  2. Contents mentioned in lines 112, 113 are contradictory to the inclusion criteria. The patients who had already undergone scoliosis correction surgery were selected for this study. Then how come the quality of life was evaluated pre-operatively? Does the patient already know that they are going to participate in this study?
  3. Line 116: Who translated the questionnaire from English to Romanian? Was the translation validated?
  4. Line 161: Follow-up was every 6 months for at least 2 years. When was the post-operative data exactly collected? By the end of 2nd year or every 6 months?

Author Response

Respected Authors,

Comment1: I am glad that you have done some corrections. However, the updated manuscript still lacks essential information. Need for the study, validation on questionnaire translation, details on follow-up data collection, and support for the conclusion are not provided or addressed.

I feel that the following points were not properly addressed or discussed in the manuscript.

The need for conducting this study was not properly discussed and justified. There are several other papers which are already been published. How far this paper is different from others?

Response1: In Romania, such studies were never conducted, even if spinal surgery is done ever since 1980. One of the aims of this study was to compare our results with the literature.

Comment 2: Contents mentioned in lines 112, 113 are contradictory to the inclusion criteria. The patients who had already undergone scoliosis correction surgery were selected for this study. Then how come the quality of life was evaluated pre-operatively? Does the patient already know that they are going to participate in this study?

Response2: Dear editor, we changed line 87 and stated: "proposed for." After the patients were proposed for surgery, they were invited to participate in the survey. We corrected the sentence as seen in lines 68-69. It was a prospective study. Otherwise, the QoL is routinely evaluated pre-operatively using SRS-30 among candidates for spinal fusion. All patients routinely agreed to participate in this study as a part of their informed consent.

Comment 3 Line 116: Who translated the questionnaire from English to Romanian? Was the translation validated?

Response 3: The questionnaire was translated and validated by the English Department from our Faculty of Medicine.

Comment 4 Line 161: Follow-up was every six months for at least two years. When was the post-operative data exactly collected? By the end of 2nd year or every six months?

Response 4: Post-op data evaluation was done in 2018 when the patients came for the regular six-month follow-up, thus having the postop evaluation done at 4,32 years after surgery.

Thank you for your comments

Reviewer 2 Report

Dear

Thank you for asking me to review the manuscript “ Clinical outcomes in patients with operated children and adolescent idiopathic scoliosis using the SRS 30 score”

This manuscript is interesting because understand which outcomes are the best predictive factors to improve the management of the patients with AIS scoliosis is one important part of our work;

Comments:

  • The author define AIS scoliosis with 3 groups base on age: under 11 years,12-14 years and 15-17 years.

Nevertheless AIS scoliosis is a pathology of adolescent. When scoliosis occurs in younger children, it can be EARLY onset scoliosis . This is not define in the manuscript and it can be different in the management. Moreover the children under 10 years haven’t the same feelings for their pathology, and the self image is not the same. Please discuss this point.

For the results, the same problem occurs. Thus the analysis associated all the kind of patients, early onset and AIS scoliosis. It can bias the results because the preoperative evaluation could not be the same. Please explain these results in each group of patients.

  • Pain is one of the major point for the follow up of these patients. In the SRS 30 score it is analyzed with severous items. Does the author have data about the anesthetic management which can modified the pain in the post operative times? if all the patients have the same kind of surgery, we don’t have any information about the anesthetic and analgesic management. Please discuss this point.
  • In the results, authors gave the results in globality. It seems interesting to have information about the patients for whom items is less or not improved after surgery in terms of

pain or self image. The globality of the results are very interesting but it is difficult to conclude without results in these kind of patients

  • The author explain that all the patients have the same kind of surgery.

Informations about gibbectomy or number of osteotomies are missing. These factors can modified the self image because when it is indicated it seems that patients have more stressed by their pathology. Please discuss.

  • The part on statistical analysis is very long without any relevant excepted for the number of subject to include. Nevertheless it seems that to have more than 48 patients ( 49) in this case they include early onset scoliosis which is a major bias. Please discuss.

Author Response

Dear

Thank you for asking me to review the manuscript "Clinical outcomes in patients with operated children and adolescent idiopathic scoliosis using the SRS 30 score"

This manuscript is interesting because understand which outcomes are the best predictive factors to improve the management of the patients with AIS scoliosis is one important part of our work;

Comments:

Comment 1: The author define AIS scoliosis with 3 groups base on age: under 11 years,12-14 years and 15-17 years.

Nevertheless AIS scoliosis is a pathology of adolescent. When scoliosis occurs in younger children, it can be EARLY onset scoliosis . This is not define in the manuscript and it can be different in the management. Moreover the children under 10 years haven't the same feelings for their pathology, and the self image is not the same. Please discuss this point.

Response1: Dear editor, thank you for your remark. We stated in the article that our evaluation was done on children and adolescents who have idiopathic scoliosis. Patients that were in groups 9-11, even a few, were girls who already had menarche and reached a Risser score of >=3, thus counting for the children with idiopathic scoliosis and not in the early onset. If needed, we can introduce this aspect in the article. "SRS-30 questions are independent of age or socioeconomic status." - lines 151-152. Every patient completed the survey under the surveillance of their parents.

Comment 2: For the results, the same problem occurs. Thus the analysis associated all the kind of patients, early onset and AIS scoliosis. It can bias the results because the preoperative evaluation could not be the same. Please explain these results in each group of patients.

Response2: Dear editor, we have provided additional data on preoperative SRS 30 scores based on age and domains. We hope this improvement will meet your expectations. Lines 206-210.

Comment 3 : Pain is one of the major point for the follow up of these patients. In the SRS 30 score it is analyzed with severous items. Does the author have data about the anesthetic management which can modified the pain in the post operative times? if all the patients have the same kind of surgery, we don't have any information about the anesthetic and analgesic management. Please discuss this point.

Response 3: Every patient received the same anesthetic and pain management amount. The anesthetic team has a protocol for the perioperative pain management used in every case of spinal fusion. We can include the list and dosing of every substance used if needed.

Comment 4:In the results, authors gave the results in globality. It seems interesting to have information about the patients for whom items is less or not improved after surgery in terms of

pain or self image. The globality of the results are very interesting but it is difficult to conclude without results in these kind of patients

Response4: Dear editor, we added Table 4 comparing the age group and the scores obtained in every domain of the SRS 30 questionnaire before and after the surgery. The additional table should help conclude the results we presented. Lines 206-210.

Comment 5: The author explain that all the patients have the same kind of surgery.

Informations about gibbectomy or number of osteotomies are missing. These factors can modified the self image because when it is indicated it seems that patients have more stressed by their pathology. Please discuss.

Response 5: In our cases, we did not perform gibbectomy or correction osteotomies.

Comment 6: The part on statistical analysis is very long without any relevant excepted for the number of subject to include. Nevertheless it seems that to have more than 48 patients ( 49) in this case they include early onset scoliosis which is a major bias. Please discuss.

Response 6: Early-onset and children / adolescent idiopathic scoliosis were addressed above. We feel that our results are concise to sustain the conclusions drawn.

Thank you for your comments 

Reviewer 3 Report

English writing must be revised. SRS-30 is a practical tool to measure conditions in spine pathology, and successful scoliosis correction needs multiple factors. author jumped into conclusion without effective evident support.

Author Response

Comment 1 : English writing must be revised. SRS-30 is a practical tool to measure conditions in spine pathology, and successful scoliosis correction needs multiple factors. author jumped into conclusion without effective evident support.

Response 1: Thank you for the review. We revised English writing. As shown in the Results, our conclusions are backed by statistical analysis, which comprised correlations between SRS-30 score, Cobb angle correction rate, and age at surgery.

Round 2

Reviewer 1 Report

Respected Authors,

Kindly incorporate your responses in the manuscript. It shall provide more clarity. The title doesnt seem appropriate to the current manuscript. Consider rephrasing it. Add more relevant texts with appropriate citations in the introduction about the need for this study. In figure 1, legends are missing. In line 291, 293, it says figure 9. Starting from the line 290, the texts seem to be more concentrated only on statistical analysis, rather than discussing about the study results. I request the authors to simplify it. This study used only SRS-30 score as a tool to evaluate the patients who underwent scoliosis correction surgery. But the conclusions are very tall. Rephrase it. 

Regards.

Author Response

Respected Authors,

Kindly incorporate your responses in the manuscript. It shall provide more clarity. 

RE: Reponses were added in lines 61-63, lines 107-108, lines 110-111,  lines 157-159.

The title doesnt seem appropriate to the current manuscript. Consider rephrasing it. 

RE: We rephrase it into: "Quality of life evaluation using SRS-30 score for operated children and adolescent idiopathic scoliosis". We hope that the title meets your expectations.

Add more relevant texts with appropriate citations in the introduction about the need for this study. 

RE: We added in lines 61-63, lines 81-83, lines 86-87. We can expand the introduction if needed even more.

In figure 1, legends are missing. 

RE: Legends were detailed as seen in lines 171-175.

In line 291, 293, it says figure 9. 

RE: It was a type-o, the text is about the last figure that talks about the power study. It's Figure 3.

Starting from the line 290, the texts seem to be more concentrated only on statistical analysis, rather than discussing about the study results. I request the authors to simplify it. This study used only SRS-30 score as a tool to evaluate the patients who underwent scoliosis correction surgery. 

RE: lines 309-338 were removed as requested. Lines 305-308 along with Figure 3 should cover all about the power analysis results of our study.

But the conclusions are very tall. Rephrase it. 

Regards.

RE: Thank you for reviewing our article and we hope that now it meets your expectations.

This manuscript is a resubmission of an earlier submission. The following is a list of the peer review reports and author responses from that submission.

Round 1

Reviewer 1 Report

Abstract:

  1. Remove numbering between each heading of the abstract
  2. Write full form of SRS-30
  3. Rather than starting sentence using numbers, I suggest you start the sentence with words.
  4. In line number 19, write the abbreviation AIS instead of adolescent idiopathic scoliosis.
  5. Can you please write the reference that the age group 9 to 17 year is of adolescent, as far as I know age group between 13 to 19 is of adolescence and age group below 12 is of children.

Introduction:

  1. Line number 41, remove the sentence, “Though the subject of our study is AIS”
  2. I think you have to write explicitly about the A.I.S, what is AIS, what are the sign and symptoms, what is the high-risk group of AIS, what is the therapeutic and management approaches for AIS, and what can be the consequences of AIS on one’s health?
  3. The rational of this study is not properly defined, although the objectives are very clear. You need to describe why you used SRS-30 and why HRQoL for assessing the clinical outcomes? How will this research help clinicians, academicians and researchers in future?

Methodology:

  1. In line number 67, write the abbreviation AIS instead of adolescent idiopathic scoliosis.
  2. SRS-30 questionnaire was used to assess the difference in the quality of life of patients before and after surgery, but this is not clear. Is it possible for you to describe explicitly how many days before and after surgery SRS-30 questionnaire was used to assess the quality of life of patients?
  3. The original SRS-30 questionnaire in English language was translated into local language Romanian. At one end, it is a good indicator if the research team had tested the validity and reliability of Romanian language version. If in case, the validity and reliability was not tested due to any reason, I advised you to write this in study limitations.
  4. I think you should describe all six Lenke classification in your methodology section
  5. In line 100, what does [x] mean?

Results:

  1. Remove the word S.D. after ± from line 102, and 103.
  2. Figure-1: Rather than presenting simple bar graph. I suggest you present box plot, because the boxplot better describe the mean and standard deviations
  3. Check the labeling of figure-2 and figure-3. I suggest you present you results in tabular forms rather than presenting graphically. Show all the mean, standard deviation, and p-value.
  4. Present findings of figure 4 and figure 5 in tabular form
  5. Line number 153 is not clear. Do you mean pre and post difference or between Lenke classification level 1 and level 3.
  6. Line number 163 to 170, you had presented the mean but standard deviation was not. Please present standard deviation together with the mean.
  7. In the methodology and abstract you have mentioned that children aged from 9 years were included, while the table-1 presented minimum age of 10 year. The class interval between three age categories was not uniform. 10-12 and 13-15 year has class interval of 3 and in last category the class interval was of 2 year? If the minimum age was of 9 year then make three categories and each with class interval of 3 year. Otherwise you can create an extra class or two class categories either with two or four class interval.

Discussion:

  1. Move some of the initial paragraphs of discussion section into the introduction section after some edits. Paragraph 3 and 4.
  2. I read the discussion but I found it is not comparing and contrasting the study findings with other studies. I suggest you compare your study findings with other study finding and to increase the external validity. If there is any difference then write the reasons.
  3. I think there is a need to write a short paragraph related to limitation of study at the end of discussion before conclusion.

Reviewer 2 Report

Respected Authors,
After reading this manuscript, I felt that the literature review was not done properly before
conducting this study. There are several research articles already available that discuss about
the lifestyle of a post-operated scoliosis patient after several years of follow-up. Below are the
comments on this manuscript.

General

1. Format & alignment consistency is missing.

2. Few grammatical errors are present.

3. More detailed descriptions about this study, the method followed, discussion of results
concerning the literature should be provided.

Introduction

1. What is the need for conducting this study? How do your results differ from the
literature?

2. Unnecessary self-citations.

3. The need for conducting this study was not properly discussed and justified.

4. Line 61: Imagistic preop data. What does it mean?

Materials and Methods

1. Contents mentioned in lines 79, 80, 81 are contradictory to the inclusion criteria. If the
patients who already underwent scoliosis correction surgery were selected for this
study, how come the quality of life was evaluated pre-operatively? Do they already
know that they are going to participate in your study?

2. Data from electronic files. What does it mean? What are those data?

3. The number of data collected and the age group of the subjects were not mentioned.

4. Line 83: Who translated the questionnaire from English to Romanian? Was the
translation validated?

Results

1. Line 106: Follow-up was every 6 months for at least 2 years. When was the post-
operative data exactly collected? By the end of 2nd year or every 6 months?

2. Figures are not clear and not easy to understand.

3. Table 1: It is supposed to be 9-12. Not 10-12.

Discussion

1. Could have done better.

2. Figures 7 & 8, its descriptive texts are unnecessary.

Conclusion

1. On what basis do the authors come to the conclusion mentioned in lines 307 & 308?